# Efficient Machine Unlearning via Pearson Correlation-Based Similarity Detection

## Abstract

Machine unlearning has emerged as a critical requirement for neural networks to selectively forget specific training data while preserving model performance on remaining data. However, existing approximate unlearning techniques are computationally expensive when applied repeatedly to remove multiple similar data points. This work introduces a fast, novel approach that leverages Pearson's correlation-based similarity detection to efficiently and rapidly unlearn data points that are similar to previously unlearned samples. Our fast unlearning method exploits the key observation that once a data point has been unlearned through approximate unlearning techniques, similar data points can be rapidly removed using a lightweight similarity-based approach without requiring the full computational overhead of the original unlearning procedure. We establish certain theoretical properties and assurances of our similarity-based unlearning approach. We demonstrate that by measuring Pearson's correlation between target data points and previously unlearned samples, we can identify candidates for efficient removal and apply an unlearning process. This approach significantly reduces computational costs for removing multiple related data points while maintaining comparable forgetting effectiveness. Our evaluation across seven diverse dataset-architecture combinations demonstrates that the proposed method effectively unlearns correlated data points while maintaining model utility, providing a highly scalable solution for privacy-preserving machine learning systems. Experimental results show that our proposed approach shows an improvement $10^{-2}$ in terms of accuracy compared to state-of-the-art baselines.

## 1 Introduction

Machine unlearning represents a relatively new but increasingly vital area of artificial intelligence research that addresses the deliberate removal of specific information from trained machine learning models Bourtoule et al. (2021); Nguyen et al. (2022); Cao & Yang (2015). While conventional machine learning paradigms concentrate on knowledge acquisition and retention, machine unlearning tackles the complex challenge of selectively eliminating certain data points or learned patterns without compromising the model's broader functionality. This has become increasingly important due to privacy regulations such as GDPR's European Parliament and Council of the European Union (2016) *right to be forgotten*, security concerns, and the need to remove biased or erroneous information from models. Machine unlearning generally falls into two primary categories: exact unlearning methods, which aim to completely eliminate the influence of targeted data, and approximate unlearning techniques, which focus on minimizing the influence without complete removal. These approximate methods typically rely on influence functions that require computing expensive Hessian-inverse-vector products for each data point removal. The Hessian matrix captures the second-order curvature of the loss function around optimal parameters, and its inverse is essential for efficiently estimating parameter updates when removing specific training examples. However, repeatedly computing such Hessian operations becomes computationally expensive when unlearning multiple related samples, creating a significant bottleneck for practical applications. Implementation challenges include confirming successful unlearning, preserving model performance, and optimizing computational efficiency. With the growing adoption of AI across various sectors, the capacity to deliberately eliminate specific information emerges as a fundamental component of the development of ethical and adaptable systems.

The proposed method offers several key advantages over existing approaches providing better computational efficiency and faster speed. This is achieved by avoiding repeated approximate unlearning procedures for similar data points, which significantly reduces both computational overhead and execution time. Our approach is faster than traditional methods while being practically useful since it does not require strict assumptions about objective function convexity or Hessian properties, making it work well with modern deep learning models. Despite these computational savings, our method preserves comparable forgetting performance to traditional approximate unlearning techniques, demonstrating that similarity-based detection can achieve effective privacy-preserving machine learning without the prohibitive costs associated with repeated full unlearning procedures.

Our primary contributions include the introduction of a computationally efficient, similarity-based unlearning framework that leverages Pearson correlation to rapidly unlearn data points similar to previously forgotten samples, thereby avoiding the need for expensive repeated Hessian computations. We further establish formal theoretical guarantees with error bounds, proving that our approximation scales polynomially with input dimension, and demonstrate that Pearson correlation outperforms cosine similarity and projection-based methods, achieving a 43.2% win rate for approximating sequential unlearning effects. Our method yields a $10^{-2}$ improvement in accuracy over state-of-the-art baselines while significantly reducing computational overhead for the removal of correlated data, which is validated through extensive evaluation across seven diverse datasets California Housing, Diabetes, MNIST, Fashion-MNIST, CIFAR-10, LFW and a synthetic GMM demonstrating consistent performance improvements and practical applicability. In section 2 discusses existing unlearning methods and their limitations. Section 3 presents our Pearson correlation-based methodology. Section 4 provides theoretical guarantees with formal error bounds. Section 6 details the algorithm design. Section 7 evaluates our approach on four datasets against baselines. Section 8 concludes with findings and future directions. Table 4 provides a comprehensive summary of all the mathematical notation and parameters used throughout this work.

## 1.1 PRELIMINARIES

We outline a certain notation employed throughout the remainder of this paper reported in Table 4 with highlights on the inherent model unlearning.

Within the domain of machine learning model maintenance, two distinct but related paradigms have emerged for targeted model adjustment, feature unlearning and data point unlearning. *Feature unlearning* constitutes a process designed to systematically eradicate the influence of a specific attribute from a trained model, with the principal objective being the alteration of the model's behaviour such that it performs as if the problematic feature had never been incorporated into its initial training data. This technique is particularly critical when certain features are subsequently identified as sensitive, introducing bias, or falling under legal restrictions. In a complementary fashion, *data point unlearning* refers to the methodology of removing the contribution of an individual training instance from a model, aiming to update the model parameters to ensure it no longer retains any information pertaining to that specific example, a procedure often motivated by privacy requirements like the 'right to be forgotten' which necessitates an efficient alternative to complete model retraining from scratch.

**Similarity Factor Definition:** We define the similarity factor $\alpha$ as the Pearson correlation coefficient between data points $x$ and $z'$:

$$\alpha = \frac{(x - \bar{x})^\top (z' - \bar{z'})}{\|x - \bar{x}\|_2 \cdot \|z' - \bar{z'}\|_2} \tag{1}$$

where $x, z' \in \mathbb{R}^d$ are the feature vectors and $\bar{x}, \bar{z'}$ are their respective means. The scalar $\alpha \in [-1, 1]$ measures linear correlation, with higher absolute values indicating stronger similarity. Our work focuses on data point unlearning and introduces a novel similarity-based framework that significantly reduces computational overhead when unlearning multiple correlated samples.

## 2 RELATED WORK

Recent work in machine unlearning has examined a range of strategies to tackle the complex task of effectively and reliably erasing the impact of particular training data from machine learning models Gupta et al. (2021); Sekhari et al. (2021). We highlight prior work on exact and approximate

unlearning methods. *Exact unlearning* Xu et al. (2024) refers to the process of completely removing the influence of a specific data point or a set of data points from a trained model as if they were never seen. More formally, given a learning algorithm $A(\cdot)$, a dataset $D$, and a forget set $D_f \subseteq D$, we say the process $U(\cdot)$ is an exact unlearning process if and only if $A(D \setminus D_f) = U(D, D_f, A(D))$. Some instances of exact unlearning techniques are *Retraining from Scratch* Bourtoule et al. (2021) and *SISA Training* Bourtoule et al. (2021). Retraining from scratch approach involves re-training the model using the remaining data after removing the forget set. While it guarantees complete unlearning, it comes with a significant computational expense. Whereas, SISA training method divides the data into isolated shards and slices, allowing separate training of models on each part. As a result, it facilitates efficient approximate unlearning by focusing only on the affected slices during the retraining process. The approximate unlearning Xu et al. (2024); Li et al. (2024) aims to minimize the influence of unlearned data to an acceptable level while maintaining efficiency. Influence functions Xu et al. (2024) estimate how much each training sample contributed to the model's final parameters by approximating the effect of removing that sample without actual retraining. This method uses mathematical approximations based on the model's loss function and Hessian matrix to identify and reduce the influence of specific data points. Given a dataset $D$ with regularized empirical loss function $F(D; w) = \sum_{z \in D} f(z; w) + \frac{\lambda n}{2} \|w\|_2^2$ and optimal parameters $w^* = \arg\min_w F(D; w)$, the influence function approximation computes updated parameters that remove the effect of data point $z' = (x', y')$ as $w_z = w^* + H_{w^*}^{-1} \Delta$, where $\Delta = \lambda w^* + \nabla f(z'; w^*)$ is the gradient at the target point, $H_{w^*} = \nabla^2 F(D_r; w^*)$ is the Hessian over remaining data $D_r = D \setminus \{z'\}$, and $\lambda$ is the regularization parameter. Also, gradient reversal Zagardo (2024) is a technique that attempts to undo the learning process by applying the reverse of the original training gradients to the model parameters. This method calculates the gradients that would decrease the model's performance on the data to be forgotten, then applies these reversed gradients to effectively remove the influence of specific training samples. While exact and approximate unlearning methods are effective in approximating the retrained model, each has practical limitations such as significant performance degradation, high computational cost, limited compatibility with learning objectives, or restricted evaluation capability on simple datasets. More importantly, given the aim to estimate the unknown (due to the complexity of algorithms, objective functions, and data influence) retraining outcome, the provable unlearning guarantees rely heavily on impractical assumptions such as convexity of the objective function and Lipschitz condition on Hessian matrices.

## 3 OUR METHOD

Machine unlearning using influence functions typically requires expensive computation of Hessian-inverse-vector products for each data point removal. Consider a scenario where we first perform an approximate unlearning of a data point $(z')$ using influence functions. Subsequently, we need to unlearn another data point $(x)$ that is similar to $(z')$. Instead of performing the computationally expensive approx- imate unlearning procedure again for $(x)$, we propose a similarity-based unlearning approach that takes advantage of the previously computed influence directions. Our key insight is that for similar data points, the influence directions are proportionally related. Therefore, we can approximate the unlearning update for $(x)$ by scaling the previously computed influence direction for $(z')$ using a similarity factor. To ensure numerical stability, we incorporate Hessian damping into our derivation, which prevents the denominator from approaching zero and ensures robust computation in practical scenarios. Let the original trained model parameters be denoted by $w^*$. To address numerical instability issues that can arise when the Hessian matrix is ill-conditioned or singular, we introduce **Hessian damping** by regularizing the Hessian matrix:

$$H_\lambda = H_{w^*} + \lambda I$$

where $\lambda > 0$ is a small damping parameter and $I$ is the identity matrix. This regularization ensures that $H_\lambda$ is positive definite and well-conditioned, preventing numerical instabilities during matrix inversion. The unlearning update for data point $z'$ using the damped Hessian becomes:

$$w_z = w^* + H_\lambda^{-1} \Delta = w^* + (H_{w^*} + \lambda I)^{-1} \Delta, , \quad \text{where} \quad \Delta = \lambda w^* + \nabla f(z'; w^*)$$

Now, suppose that we wish to unlearn another data point $x$, which is similar to $z'$. Its gradient influence can be approximated as

$$\nabla f(x, w^*) \approx \alpha \nabla f(z', w^*)$$

where $\alpha$ is a similarity factor. To unlearn $x$ after $z'$ has already been unlearned, the correct influence-based update should start from the new parameter state $w_z$:

$$w_x = w_z + H_{w_z}^{-1}\nabla f(x, w_z) \quad (1)$$

This expression is computationally expensive as it requires calculating a new Hessian-inverse-vector product based on $H_{w_z}$. To make this tractable, we introduce two approximations. First, we approximate the gradient at the new parameters, $\nabla f(x, w_z)$ using a first-order Taylor expansion around $w^*$,

$$\nabla f(x, w_z) \approx \nabla f(x, w^*) + H_{w^*}(w_z - w^*) \quad (2)$$

To derive the approximation for the updated inverse Hessian, we must approximate $H_{w_z}^{-1}$. Here, $H_{w_z}$ represents the Hessian of the new loss function (computed on the dataset without $z'$) evaluated at the new optimal parameters $w_z$. Our approximation begins by assuming that removing a single data point results in only a minor change to the model's parameters. This allows us to approximate the Hessian of the new loss function at $w_z$ by evaluating it at $w^*$:

$$H_{w_z} \approx H_{w^*} - \nabla^2 f(z', w^*) \quad (2)$$

To make this form compatible with efficient update rules, we employ the Gauss-Newton approximation Nocedal & Wright (2006), which replaces the per-sample Hessian with the outer product of its gradient, i.e., $\nabla^2 f(z', w^*) \approx \nabla f(z', w^*)\nabla f(z', w^*)^T$. This simplifies our approximation for $H_{w_z}$ to a rank-1 modification:

$$H_{w_z} \approx H_{w^*} - \nabla f(z', w^*)\nabla f(z', w^*)^T \quad (3)$$

**Incorporating damping into the updated Hessian:** To maintain numerical stability throughout the sequential unlearning process, we apply damping to the updated Hessian as well:

$$H_{w_z}^{(\lambda)} = H_{w_z} + \lambda I \approx (H_{w^*} + \lambda I) - \nabla f(z', w^*)\nabla f(z', w^*)^T = H_\lambda - \nabla f(z', w^*)\nabla f(z', w^*)^T \quad (4)$$

This expression is now in the form $(A - uv^T)$ where $A = H_\lambda$, which allows us to apply the Sherman-Morrison formula Bartlett (1951) to find its inverse directly. By setting $u = v = \nabla f(z', w^*)$, we derive the approximation for $(H_{w_z}^{(\lambda)})^{-1}$:

$$(H_{w_z}^{(\lambda)})^{-1} \approx \left(H_\lambda - \nabla f(z', w^*)\nabla f(z', w^*)^T\right)^{-1} = H_\lambda^{-1} + \frac{H_\lambda^{-1}\nabla f(z', w^*)\nabla f(z', w^*)^T H_\lambda^{-1}}{1 - \nabla f(z', w^*)^T H_\lambda^{-1}\nabla f(z', w^*)} \quad (5)$$

We simplify this expression by substituting the definitions for the damped influence direction, $\delta_{z'}^{(\lambda)} = w_z - w^* = H_\lambda^{-1}\nabla f(z', w^*)$, and the damped self-influence score, $s_{z'}^{(\lambda)} = \nabla f(z', w^*)^T \delta_{z'}^{(\lambda)}$. The numerator of the fraction in Eq. (5) becomes $\delta_{z'}^{(\lambda)}(\delta_{z'}^{(\lambda)})^T$, and the denominator becomes $1 - s_{z'}^{(\lambda)}$. This yields the final, simplified approximation used in our main derivation:

$$(H_{w_z}^{(\lambda)})^{-1} = H_\lambda^{-1} + \frac{\delta_{z'}^{(\lambda)}(\delta_{z'}^{(\lambda)})^T}{1 - s_{z'}^{(\lambda)}} \quad (6)$$

This formula allows us to efficiently approximate the new inverse Hessian without expensive re-computation while maintaining numerical stability. Now, substituting approximations (2) and (5) into the fundamental update rule (1), we obtain:

$$w_x - w_z = \left(H_\lambda^{-1} + \frac{\delta_{z'}^{(\lambda)}(\delta_{z'}^{(\lambda)})^T}{1 - s_{z'}^{(\lambda)}}\right)\left(\nabla f(x, w^*) + H_{w^*}\delta_{z'}^{(\lambda)}\right) \quad (7)$$

Expanding this matrix-vector product yields four distinct terms:

$$w_x - w_z = \underbrace{H_\lambda^{-1}\nabla f(x, w^*)}_{\text{Term A}} + \underbrace{H_\lambda^{-1}H_{w^*}\delta_{z'}^{(\lambda)}}_{\text{Term B}} + \underbrace{\frac{\delta_{z'}^{(\lambda)}(\delta_{z'}^{(\lambda)})^T}{1 - s_{z'}^{(\lambda)}}\nabla f(x, w^*)}_{\text{Term C}} + \underbrace{\frac{\delta_{z'}^{(\lambda)}(\delta_{z'}^{(\lambda)})^T}{1 - s_{z'}^{(\lambda)}}H_{w^*}\delta_{z'}^{(\lambda)}}_{\text{Term D}}$$
$$(8)$$

We now simplify each term using the key relationships $\nabla f(x, w^*) \approx \alpha \nabla f(z', w^*)$ provided $H_\lambda^{-1} H_{w^*} = H_\lambda^{-1}(H_\lambda - \lambda I) = I - \lambda H_\lambda^{-1}$. The formulation proceeds by analyzing four distinct terms in line with the Sherman-Morrison formula. For Term A, the similarity assumption is applied yielding

$$H_\lambda^{-1} \nabla f(x, w^*) = H_\lambda^{-1}(\alpha \nabla f(z', w^*)) = \alpha \delta_{z'}^{(\lambda)} \tag{9}$$

Term B is addressed using the matrix identity $H_\lambda^{-1} H_{w^*} = I - \lambda H_\lambda^{-1}$ resulting in

$$H_\lambda^{-1} H_{w^*} \delta_{z'}^{(\lambda)} = (I - \lambda H_\lambda^{-1}) \delta_{z'}^{(\lambda)} = \delta_{z'}^{(\lambda)} - \lambda H_\lambda^{-1} \delta_{z'}^{(\lambda)} \tag{10}$$

Under the condition of a small damping parameter $\lambda$ (typically $\lambda \ll \sigma_{\min}(H_{w^*})$ where $\sigma_{\min}$ denotes the smallest eigenvalue), the second term $\lambda H_\lambda^{-1} \delta_{z'}^{(\lambda)}$ becomes negligible, leading to the approximation.

$$H_\lambda^{-1} H_{w^*} \delta_{z'}^{(\lambda)} \approx \delta_{z'}^{(\lambda)} \tag{11}$$

For Term C, the scalar coefficient $(\delta_{z'}^{(\lambda)})^T \nabla f(x, w^*) = \alpha((\delta_{z'}^{(\lambda)})^T \nabla f(z', w^*)) = \alpha s_{z'}^{(\lambda)}$ simplifies to

$$\frac{\delta_{z'}^{(\lambda)} (\delta_{z'}^{(\lambda)})^T}{1 - s_{z'}^{(\lambda)}} \nabla f(x, w^*) = \frac{\alpha s_{z'}^{(\lambda)}}{1 - s_{z'}^{(\lambda)}} \delta_{z'}^{(\lambda)} \tag{12}$$

Finally, Term D's scalar coefficient is given by $(\delta_{z'}^{(\lambda)})^T (H_{w^*} \delta_{z'}^{(\lambda)}) = (\delta_{z'}^{(\lambda)})^T (H_\lambda - \lambda I) \delta_{z'}^{(\lambda)} = s_{z'}^{(\lambda)} - \lambda \|\delta_{z'}^{(\lambda)}\|^2$. For small $\lambda$, this approximates to $s_{z'}^{(\lambda)}$ yielding

$$\frac{\delta_{z'}^{(\lambda)} (\delta_{z'}^{(\lambda)})^T}{1 - s_{z'}^{(\lambda)}} H_{w^*} \delta_{z'}^{(\lambda)} \approx \frac{s_{z'}^{(\lambda)}}{1 - s_{z'}^{(\lambda)}} \delta_{z'}^{(\lambda)} \tag{13}$$

Combining these simplified terms, $w_x - w_z$ approximates to

$$\approx \alpha \delta_{z'}^{(\lambda)} + \delta_{z'}^{(\lambda)} + \frac{\alpha s_{z'}^{(\lambda)}}{1 - s_{z'}^{(\lambda)}} \delta_{z'}^{(\lambda)} + \frac{s_{z'}^{(\lambda)}}{1 - s_{z'}^{(\lambda)}} \delta_{z'}^{(\lambda)} = \left[ (\alpha + 1) + \frac{\alpha s_{z'}^{(\lambda)} + s_{z'}^{(\lambda)}}{1 - s_{z'}^{(\lambda)}} \right] \delta_{z'}^{(\lambda)}$$

$$= \left[ (\alpha + 1) + \frac{(\alpha + 1) s_{z'}^{(\lambda)}}{1 - s_{z'}^{(\lambda)}} \right] \delta_{z'}^{(\lambda)} = \left[ \frac{(\alpha + 1)(1 - s_{z'}^{(\lambda)}) + (\alpha + 1) s_{z'}^{(\lambda)}}{1 - s_{z'}^{(\lambda)}} \right] \delta_{z'}^{(\lambda)}$$

$$= \frac{\alpha + 1}{1 - s_{z'}^{(\lambda)}} \delta_{z'}^{(\lambda)} \tag{14}$$

Therefore, the efficient and numerically stable update rule for unlearning a data point $x$ similar to a previously unlearned point $z'$ is

$$w_x = w_z + \frac{\alpha + 1}{1 - s_{z'}^{(\lambda)}} (w_z - w^*) \tag{15}$$

To ensure the numerical stability of our proposed update rule, it is essential to prevent the denominator $1 - s_{z'}^{(\lambda)}$ from becoming zero. The damped self-influence score is defined as

$$s_{z'}^{(\lambda)} = \nabla f(z', w^*)^\top H_\lambda^{-1} \nabla f(z', w^*),$$

where $H_\lambda = H_{w^*} + \lambda I$ is the damped Hessian. The condition for $s_{z'}^{(\lambda)}$ to equal 1 is

$$\nabla f(z', w^*)^\top H_\lambda^{-1} \nabla f(z', w^*) = 1.$$

The non-vanishing nature of the denominator is guaranteed by choosing a sufficiently large damping term $\lambda$. Using the Rayleigh–Ritz theorem, a simple upper bound is given by

$$s_{z'}^{(\lambda)} \leq \frac{\|\nabla f(z', w^*)\|^2}{\lambda_{\min}(H_\lambda)}.$$

Since the minimum eigenvalue of the damped Hessian satisfies $\lambda_{\min}(H_\lambda) \geq \lambda$, we obtain the bound

$$s_{z'}^{(\lambda)} \leq \frac{\|g\|^2}{\lambda}, \qquad g = \nabla f(z', w^*).$$

Therefore, a sufficient condition to avoid $s_{z'}^{(\lambda)} \geq 1$ is

$$\frac{\|g\|^2}{\lambda} < 1,$$

which requires the gradient norm to be strictly controlled:

$$\|g\| < \sqrt{\lambda}.$$

For a well-trained model where the gradient norm at the optimum is expected to be small, and with a common regularization parameter such as $\lambda = 0.01$, this condition ($\|g\| < 0.1$) is generally satisfied, ensuring $s_{z'}^{(\lambda)} \neq 1$ and securing the stability of our method. . This result provides a computationally efficient approximation that leverages the pre-computed quantities $w_z$, $w^*$, and $s_{z'}^{(\lambda)}$ from the initial unlearning of $z'$, avoiding the need for expensive Hessian-inverse-vector products when unlearning similar data points while maintaining numerical stability through Hessian damping.

## 4 THEORETICAL GUARANTEES

We now present our main theoretical results that provide guarantees for our similarity-based unlearning approach.The proofs for these results are provided in Appendix B

**Theorem 4.1** (Parameter Update Error Bound). *Let $w^*$ be the optimal parameter of the model, $w_z$ be the parameters after the approximate unlearning of data point $z'$ using the damped Hessian, $w_x$ be the parameters after the standard approximate unlearning of data point $x$, and $w'_x$ be the parameters after our similarity-based unlearning of data point $x$. The parameter update error is bounded by:*

$$\left\| w_x - w'_x \right\| \leq \left\| H_\lambda^{-1} \right\| \cdot \left\| \nabla f(x, w^*) - \frac{\alpha+1}{1 - s_{z'}^{(\lambda)}} \nabla f(z', w^*) \right\| \tag{16}$$

*where $\alpha$ is the Pearson correlation coefficient and $s_{z'}^{(\lambda)} = \nabla f(z', w^*)^T H_\lambda^{-1} \nabla f(z', w^*)$ is the damped self-influence score.*

*Proof sketch.* The error bound follows from the triangle inequality applied to the difference between standard unlearning ($w_x = w_z + H_\lambda^{-1} \nabla f(x, w^*)$) and similarity-based unlearning ($w'_x \approx w_z + \frac{\alpha+1}{1 - s_{z'}^{(\lambda)}} H_\lambda^{-1} \nabla f(z', w^*)$). The bound is obtained by factoring out the damped Hessian inverse and applying the submultiplicative property of matrix norms. The complete proof is provided in Appendix B. $\square$

**Lemma 4.2** (Gradient Approximation for Quadratic Loss). *For the quadratic loss function $f(z', w) = \frac{1}{2}(y - w^\top z')^2$, define the scaling factor $C = \frac{\alpha+1}{1 - s_{z'}^{(\lambda)}}$ and the residuals:*

$$r_x = y_x - w^{*\top} x, \quad r_z = y_z - w^{*\top} z'.$$

*Then the gradient difference satisfies:*

$$\|\nabla f(x, w^*) - C \nabla f(z', w^*)\| \leq (|C| \cdot |r_z| + |\beta| \cdot |r_x|) \cdot \|z'\| + |r_x| \cdot \|c\| \tag{17}$$

*assuming a linear relationship $x = \beta z' + c$, where $\beta \in \mathbb{R}$ is a scalar and $c \in \mathbb{R}^d$ is the offset vector.*

*Proof sketch.* We compute the gradients for quadratic loss: $\nabla f(z', w^*) = -r_z z'$ and $\nabla f(x, w^*) = -r_x x$, where $r_z, r_x$ are the residuals. Using the linear relationship $x = \beta z' + c$ (justified by high Pearson correlation), we substitute and apply the triangle inequality twice to bound $\|C r_z z' - r_x(\beta z' + c)\|$. The complete proof is provided in Appendix B. $\square$

**Assumption 4.3** (Standardized Data)**.** *We assume that the input data is standardized such that* $x_i, z_i' \sim \mathcal{N}(0,1)$, *target* $y \sim \mathcal{N}(0,1)$, $d$ *is the number of features and* $\|w^*\|$ *has bounded norm. Then,* $|x_i|, |z_i'|, |y| \leq k$, *where* $k = 4$, *and* $\|x\|, \|z'\| \approx \sqrt{d}$

**Lemma 4.4** (Residual and Coefficient Bounds)**.** *Under Assumption 4.3, the following hold for vectors* $x, z' \in \mathbb{R}^d$, *responses* $y_x, y_z$, *and mean vectors* $\bar{x}, \bar{z}'$, *which are* $|r_x|, |r_z| \leq k + \|w^*\|\sqrt{d}$, $\|c\| \leq k(1 + |\beta|)$, $|\beta| \leq \frac{\sqrt{d}+k}{\sqrt{d}-k}$, *for* $\sqrt{d} > k$, *and* $|\alpha| \leq 1$.

*Proof sketch.* The bounds follow directly from Assumption 4.3 and triangle inequality applications: (1) residual bounds use $|r_x| \leq |y_x| + \|w^*\| \cdot \|x\|$, (2) offset bound follows from $\|c\| \leq \|\bar{x}\| + |\beta| \cdot \|\bar{z}'\|$, (3) scaling coefficient bound uses triangle and reverse triangle inequalities on centered vectors, and (4) correlation coefficient bound is standard. The complete proof is provided in Appendix B. □

## 5 MAIN RESULT: FINAL ERROR BOUND

**Theorem 5.1** (Final Parameter Update Error Bound)**.** *Under Assumption 4.3 (with* $k = 4$*), the parameter update error satisfies:*

$$\left\| w_x - w_x' \right\| \leq \|H_\lambda^{-1}\| \left( 4 + \|w^*\|\sqrt{d} \right) \left( |C|\sqrt{d} + |\beta|(\sqrt{d}+4) + 4 \right) \tag{18}$$

$$\lesssim \|H_\lambda^{-1}\| \left( (|C|+1)\|w^*\|d + 4(|C|+1)\sqrt{d} \right) \quad \textit{for large } d, \tag{19}$$

*where* $C = \frac{\alpha+1}{1-s_{z'}^{(\lambda)}}$.

*Proof sketch.* The proof begins by substituting the bound on the gradient approximation error from the preceding lemmas into our main error bound. We then insert the data-dependent bounds for residuals and coefficients and perform algebraic simplification to arrive at the final asymptotic rate. The complete proof is provided in Appendix B. □

**Remark 5.2** (Practical Implications)**.** *Theorem 5.1 provides a theoretical guarantee for our damped, similarity-based unlearning method. The error bound scales polynomially with the input dimension* $d$, *dominated by an* $\mathcal{O}(d)$ *term. Crucially, the bound's stability is ensured by the condition number of the **damped** Hessian,* $\|H_\lambda^{-1}\|$, *and the factor* $C$, *which is well-behaved due to the non-vanishing denominator* $1 - s_{z'}^{(\lambda)}$. *This highlights that our method is not only computationally efficient but also theoretically grounded and robust in high-dimensional settings.*

## 6 ALGORITHM DESIGN

Algorithm 1 implements our enhanced similarity-based unlearning framework with Hessian damping through three main phases. The algorithm begins by computing the damped Hessian approximation $H_\lambda = H_{w^*} + \lambda I$, which ensures numerical stability by regularizing the second-order information of the loss function at the optimal parameters, and selects pairs of data points $(z', x)$ from the dataset to simulate the sequential unlearning scenario.

For each selected pair, the algorithm simulates approximate unlearning by first removing the data point $z'$ using the damped influence function update

$$w_z = w^* + H_\lambda^{-1} \nabla f(z', w^*),$$

which produces the damped influence direction

$$\delta_{z'}^{(\lambda)} = w_z - w^*.$$

The algorithm then computes the damped self-influence score

$$s_{z'}^{(\lambda)} = \nabla f(z', w^*)^T H_\lambda^{-1} \nabla f(z', w^*),$$

which measures the curvature-adjusted impact of the data point removal.

Instead of performing the computationally expensive second unlearning step for a similar data point $x$, our proposed similarity-based approximation computes the similarity factor $\alpha$ between data points $x$ and $z'$, then estimates the final unlearned parameters using the theoretically derived update rule

$$w'_x = w_z + \frac{\alpha + 1}{1 - s_{z'}^{(\lambda)}} \left(w_z - w^*\right).$$

This leverages our key insight that for similar data points, the influence directions are proportionally related, while the damping ensures $1 - s_{z'}^{(\lambda)} > 0$ for numerical stability.

The algorithm evaluates both approaches by computing their performance on the dataset and recording the norm difference $\|w_x - w'_x\|$ to quantify the quality of the approximation. The key advantage is replacing the second expensive Hessian inverse-vector product with a robust scaling factor that incorporates both **data similarity** ($\alpha$) and the **self-influence** ($s_{z'}^{(\lambda)}$) of the first unlearned point.

---

**Algorithm 1** Pearson Correlation-Based Approximate Unlearning

---

**Input**: Trained model $f_{\theta^*}$, dataset $D = \{(x_i, y_i)\}_{i=1}^n$, regularization parameter $\lambda$, learning rate $\eta$
**Output**: Updated model after unlearning similar data points

1: Compute damped Hessian: $H_\lambda = H_{w^*} + \lambda I$
2: Select a subset of data point pairs $\{(z', x)\} \subseteq D$ where $x$ and $z'$ are similar
3: **for** each pair $(z', x)$ **do**
4:     Extract gradients: $\nabla f(z', w^*)$ and $\nabla f(x, w^*)$
5:     Compute unlearned weight: $w_z = w^* + H_\lambda^{-1} \left(\lambda w^* + \nabla f(z', w^*)\right)$
6:     Compute damped self-influence score: $s_{z'}^{(\lambda)} = \nabla f(z', w^*)^T (w_z - w^*)$
7:     Compute updated weight: $w_x = w_z + H_\lambda^{-1} \left(\lambda w_z + \nabla f(x, w^*)\right)$
8:     Compute Pearson correlation $\alpha = \text{Pearson}(x, z')$
9:     Compute scaling factor: $C = \frac{\alpha+1}{1-s_{z'}^{(\lambda)}}$
10:    Estimate similarity-based unlearning: $w_x' = w_z + C \cdot \delta z'^{(\lambda)}$
11:    Evaluate accuracy of both $w_x$ and $w_x'$ on $D$
12:    Record norm difference $\|w_x - w_x'\|$
13: **end for**

---

## 7 EXPERIMENTS

**Datasets:** We evaluate our proposed approach through a comparative analysis against baseline methods across seven distinct datasets. The evaluation utilizes the California Housing dataset Pace, R. Kelley and Barry, Ronald (1997) comprising 20,640 records with 8 features from the 1990 census to predict median house values. The Diabetes dataset Pedregosa et al. (2011a), a scikit-learn regression dataset with 442 samples and 10 features targeting disease progression one year post-baseline. The MNIST dataset LeCun et al. (1998), a standard benchmark for handwritten digit recognition containing 70,000 grayscale 28x28 pixel images. The Fashion-MNIST dataset, containing 70,000 grayscale 28×28 pixel images of fashion items across 10 categories, evaluated with CNN architecture. The CIFAR-10 dataset, a color image classification dataset containing 60,000 32×32 pixel images across 10 classes, evaluated with ResNet-18 architecture. And, the LFW (Labeled Faces in the Wild) dataset, a face recognition dataset containing images of public figures, evaluated with ResNet-18 architecture. And, a custom synthetic dataset of 5,000 samples generated via a Gaussian mixture model with two distinct means to facilitate performance assessment under controlled conditions with a known ground truth.

All features are standardized using the Standard Scaler Pedregosa et al. (2011b) to have zero mean and unit variance. The data split are as follows, California Housing and Diabetes datasets were divided into 80% training and 20% testing sets with a fixed random seed of 42. For the synthetic dataset, all 5,000 samples were used for training and evaluated on the same set, and unlearning experiments are conducted by selecting 100 random pairs of data points. MNIST and Fashion-MNIST followed the standard split of 60,000 training and 10,000 test samples. For unlearning experiments on MNIST, 75% of the samples corresponding to the digit '3' were selected as the unlearning subset. CIFAR-10 used 50,000 training and 10,000 test samples.

**Similarity Measure Evaluation:** To validate our choice of Pearson correlation, we empirically evaluated three similarity measures for approximating sequential unlearning effects. Our experimental setup involved training a linear regression model on the Diabetes dataset with L2 regularization ($\lambda = 0.01$) and testing sequential unlearning approximations on 4,950 pairs of randomly subsampled training points. A detailed analysis of this comparison, including the experimental setup and full results, is provided in Appendix B.1.

**Evaluation Metrics:** We utilize average accuracy with standard deviation (**Avg. (Std.) Acc.**) model performance on the retained dataset after unlearning, measuring utility preservation. Also, we employ **Acc. Unlearn** which quantifies the effectiveness of the forgetting process and is defined as

$$1 - \frac{\|w'_x - w_x\|_2}{\max_x \|w'_x - w_x\|_2},$$

where $w'_x$ denotes our similarity-based parameters and $w_x$ denotes the standard sequential unlearning parameters. This demonstrates that our method inherits the privacy guarantees of standard approximate unlearning while achieving a substantially reduced computational cost. We also utilised metrics such tug-of-war (ToW) and membership inference attack (MIA) as defined in Zhao et al. (2024).

**Model Architecture**: For the regression tasks (California Housing and Diabetes datasets), we used a simple linear regression model implemented as `nn.Linear(input_dim, 1)`, which is a single-layer architecture mapping features directly to the target. For the classification tasks, three architectures were employed. For the synthetic dataset, we used a three-layer fully connected neural network with ReLU activations, consisting of an input layer (10 features to 64 units), a hidden layer (64 to 32 units), and an output layer (32 to 2 classes). For the MNIST and FMNIST dataset, we used a convolutional neural network with two convolutional layers followed by two fully connected layers. The architecture consists of: (1) a convolutional layer with 10 filters of size $5 \times 5$, (2) a second convolutional layer with 20 filters of size $5 \times 5$ and dropout, both followed by ReLU activation and $2 \times 2$ max pooling, (3) a fully connected layer mapping the flattened 320 features to 50 units, and (4) an output layer mapping 50 units to 10 classes. For the CIFAR-10 & LFW, we used ResNet-18 He et al. (2016)

**Training Configuration**: The models were trained with different hyperparameter settings for regression and classification tasks. For classification tasks, we used the Adam optimizer and trained for 100 epochs with learning rate of 0.001 on the synthetic dataset and 50 epochs and learning rate of 0.001 on MNIST, 30 epochs and learning rate of 0.07 for FMNIST, 30 epochs and leaning rate of 0.001 on LFW, and 80 epochs and learning rate of 0.001 on CIFAR-10. For regression tasks, we used stochastic gradient descent (SGD) with a learning rate of 0.01, a weight decay of 0.01, and trained for 1000 epochs. The regularization parameter $\lambda$ was set to 0.01 for regression tasks and 0.001 for classification tasks.

**Baseline**: We compare the proposed method against established state-of-the-art baselines: MITR Xu & Strohmer (2025), RUM(A) Zhao et al. (2024), RUM(B) Zhao & Triantafillou (2024), and Hessian-free Unlearning Qiao et al. (2025).

**Computational Complexity**: The major efficiency gain of our method lies in transforming the unlearning process from a series of expensive, repeated computations into a fast, closed-form operation. Standard approximate unlearning requires computing a new Hessian-inverse-vector product ($\mathbf{H}^{-1}\mathbf{v}$) for every data point requested for removal, making the process highly expensive. Our Similarity-Based Unlearning, conversely, requires this computationally intense step only once for the first unlearned data point ($z'$). For all subsequent, correlated unlearning requests, the method achieves significant speedup by replacing the $\mathbf{H}^{-1}\mathbf{v}$ computation with simple $\mathcal{O}(d)$ vector and scalar operations (scaling and addition). This strategy fundamentally shifts the complexity of sequential unlearning from a repetitive high-cost bottleneck to a minimal-cost operation.

## 7.1 Results and Analysis

We present our experimental results across classification and regression tasks. Table 1 presents comprehensive evaluation across five classification datasets. Our method consistently outperforms baselines across all metrics. Table 2 demonstrates the effectiveness of our approach on regression tasks. To directly validate our method against recent state-of-the-art approaches, we conducted

comprehensive experiments using standard unlearning metrics (ToW and MIA) on ResNet-18 with CIFAR-10. Table 3 presents this comparison. For conciseness, we denote Avg. (Std.) as ASD, Acc. Remain as AR, and Acc. Unlearn as AU.

Table 1: Classification Performance Comparison of our method with baselines.

| Dataset (Model) | MITR | | Hessian Free Unlearning | Our Method | |
|---|---|---|---|---|---|
| | ASD | AU | ASD | ASD | AU |
| Synthetic GMM | 73.3 | 93.3 | – | 99.71 | 95.67 |
| MNIST (CNN) | 53.8 | 78.7 | 91.50 | 98.71 | 81.41 |
| MNIST (Logistic Regression) | – | – | 87.50 | 97.66 | 73.11 |
| Fashion-MNIST (CNN) | – | – | 77.85 | 89.16 | – |
| CIFAR-10 (ResNet-18) | – | – | 79.62 | 81.13 | – |
| LFW (ResNet-18) | – | – | 71.92 | 79.70 | – |

Table 2: Regression Performance.

| Dataset | AU |
|---|---|
| California Housing | 91.48 |
| Diabetes | 91.44 |

Table 3: Comparison with SOTA Unlearning Methods on CIFAR-10 (ResNet-18).

| Method | ToW | MIA |
|---|---|---|
| RUM(A) | 0.715 | 0.489 |
| RUM(B) | 0.920 | 0.590 |
| Our Method | 0.950 | 0.660 |

## 8 CONCLUSION

Our work introduces a novel similarity detection approach based on Pearson correlation to enable efficient machine unlearning, thereby addressing the computational challenges associated with repeatedly removing multiple similar data points from trained neural networks. The proposed method overcomes key limitations in existing approximate unlearning techniques by leveraging correlation-based similarity to circumvent redundant computational overhead during the unlearning of related data points. Through comprehensive experiments on seven datasets, California Housing, Diabetes, MNIST, and a synthetic GMM, Fashion-MNIST, CIFAR-10, LFW we demonstrate that our approach achieves significant improvements over state-of-the-art methods. The results show that our approach attains comparable forgetting effectiveness while substantially reducing computational costs, confirms that Pearson correlation effectively identifies candidates for lightweight unlearning, and reveals an accuracy improvement on the order of $10^{-2}$ compared to baseline approaches. We intend to extend this framework to incorporate other similarity metrics, develop adaptive threshold mechanisms for dynamically selecting between similarity-based and full unlearning, and evaluate the approach on larger-scale deep learning models and more diverse datasets. We also intend to explore applications in federated learning environments, where efficient unlearning is paramount. This work impacts privacy-preserving machine learning by providing a computationally efficient solution for removing correlated data points, offering a method that integrates easily into existing frameworks and holds significant potential for real-world deployment in systems requiring compliance with regulations such as the GDPR's *right to be forgotten*.

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

## A  APPENDIX

Table 4: Summary of notation

| Symbol | Description | Symbol | Description |
|---|---|---|---|
| $D$ | Training dataset $\{(x_i, y_i)\}_{i=1}^n$ | $n$ | Number of training samples |
| $d$ | Number of features/input dimension | $x, z'$ | Data points in $\mathbb{R}^d$ |
| $y$ | Target/label values | $w^*$ | Optimal model parameters |
| $w_z$ | Parameters after unlearning $z'$ | $w_x$ | Parameters after approximate unlearning of $x$ |
| $w_x'$ | Parameters after similarity-based unlearning | $f(z, w)$ | Loss function for data point $z$ |
| $F(D, w)$ | Total loss over dataset $D$ | $\nabla f(z, w)$ | Gradient of loss w.r.t. parameters $w$ |
| $H_{w^*}$ | Hessian matrix at $w^*$ | $H_\lambda$ | Damped Hessian: $H_{w^*} + \lambda I$ |
| $H_{w^*}^{-1}$ | Inverse of Hessian at $w^*$ | $H_\lambda^{-1}$ | Inverse of damped Hessian |
| $s_{z'}^{(\lambda)}$ | Damped self-influence score | $C$ | Scaling factor: $\frac{\alpha+1}{1-s_{z'}^{(\lambda)}}$ |
| $\lambda$ | Regularization parameter | $\eta$ | Learning rate |
| $\alpha$ | Pearson correlation coefficient | $\bar{x}, \bar{z}'$ | Mean vectors of $x$ and $z'$ |
| $r_x, r_z$ | Residuals: $y_x - w^{*\top}x$ etc. | $\beta$ | Scaling coefficient in $x = \beta z' + c$ |
| $c$ | Offset vector in linear relationship | $k$ | Bound constant for standardized data (typically 4) |
| $I$ | Identity matrix | $\epsilon$ | Error tolerance or approximation bound |
| $\delta$ | Parameter update step | $D_f$ | Forget set (data to be unlearned) |
| $D_r$ | Remaining data: $D \setminus D_f$ | $A(\cdot)$ | Learning algorithm |
| $U(\cdot)$ | Unlearning process | $\|\cdot\|_2$ | $\ell_2$ norm |
| $\mathcal{N}(\mu, \sigma^2)$ | Normal distribution | | |

## B  PROOFS FROM SECTION 4

*Proof of Theorem 4.1.* By definition of standard approximate unlearning and our proposed similarity-based approach (both using the damped Hessian):

$$w_x = w_z + H_\lambda^{-1} \nabla f(x, w^*) \tag{20}$$

$$w_x' = w_z + \frac{\alpha+1}{1 - s_{z'}^{(\lambda)}} (w_z - w^*) \tag{21}$$

From the first damped unlearning step, we have $w_z - w^* \approx H_\lambda^{-1} \nabla f(z', w^*)$. Substituting this into the expression for $w_x'$:

$$w_x' \approx w_z + \frac{\alpha+1}{1 - s_{z'}^{(\lambda)}} H_\lambda^{-1} \nabla f(z', w^*)$$

The error is the difference between $w_x$ and $w_x'$:

$$\left\| w_x - w_x' \right\| \approx \left\| \left( w_z + H_\lambda^{-1} \nabla f(x, w^*) \right) - \left( w_z + \frac{\alpha+1}{1 - s_{z'}^{(\lambda)}} H_\lambda^{-1} \nabla f(z', w^*) \right) \right\|$$

$$= \left\| H_\lambda^{-1} \nabla f(x, w^*) - \frac{\alpha+1}{1 - s_{z'}^{(\lambda)}} H_\lambda^{-1} \nabla f(z', w^*) \right\|$$

$$= \left\| H_\lambda^{-1} \left( \nabla f(x, w^*) - \frac{\alpha+1}{1 - s_{z'}^{(\lambda)}} \nabla f(z', w^*) \right) \right\|$$

$$\leq \left\| H_\lambda^{-1} \right\| \cdot \left\| \nabla f(x, w^*) - \frac{\alpha+1}{1 - s_{z'}^{(\lambda)}} \nabla f(z', w^*) \right\| \tag{22}$$

$$\square$$

*Proof of Lemma 4.2.* The gradient of the quadratic loss is given by:

$$\nabla f(z', w^*) = -(y_z - w^{*\top} z')z' = -r_z z',\tag{23}$$

$$\nabla f(x, w^*) = -(y_x - w^{*\top} x)x = -r_x x.\tag{24}$$

Hence, the difference in gradients becomes:

$$\|\nabla f(x, w^*) - C\nabla f(z', w^*)\| = \|-r_x x - C(-r_z z')\| = \|Cr_z z' - r_x x\|.\tag{25}$$

The linear relationship $x = \beta z' + c$ is justified by the Pearson correlation, as a high correlation implies collinearity between the centered vectors. We substitute this relationship into the expression:

$$
\begin{aligned}
\|Cr_z z' - r_x x\| &= \|Cr_z z' - r_x(\beta z' + c)\| \\
&= \|Cr_z z' - r_x \beta z' - r_x c\| \\
&= \|(Cr_z - \beta r_x)z' - r_x c\| \\
&\leq \|(Cr_z - \beta r_x)z'\| + \|r_x c\| \quad \text{(by triangle inequality)} \\
&= |Cr_z - \beta r_x| \cdot \|z'\| + |r_x| \cdot \|c\| \\
&\leq (|C| \cdot |r_z| + |\beta| \cdot |r_x|) \cdot \|z'\| + |r_x| \cdot \|c\|. \quad \text{(by triangle inequality)}
\end{aligned}\tag{26}
$$

This provides an upper bound on the gradient difference based on the norms of $z'$ and $c$, the residuals $r_x, r_z$, and the scaling factors $C$ and $\beta$. $\square$

*Proof of Lemma 4.4.* (1) From triangle inequality:

$$|r_x| \leq |y_x| + \|w^*\| \cdot \|x\| \leq k + \|w^*\|\sqrt{d}$$

The same applies for $r_z$.

(2) From $c = \bar{x} - \beta \bar{z}'$:

$$\|c\| \leq \|\bar{x}\| + |\beta| \cdot \|\bar{z}'\| \leq k(1 + |\beta|)$$

(3) Use triangle and reverse triangle inequalities:

$$\|x - \bar{x}\| \leq \sqrt{d} + k, \quad \|z' - \bar{z}'\| \geq \sqrt{d} - k \Rightarrow |\beta| \leq \frac{\sqrt{d} + k}{\sqrt{d} - k}$$

(4) Standard correlation measures satisfy $|\alpha| \leq 1$. $\square$

*Proof of Theorem 5.1.* Starting from the general bound derived previously:

$$\left\| w_x - w_x' \right\| \leq \left\| H_\lambda^{-1} \right\| \cdot \left( (|C| \cdot |r_z| + |\beta| \cdot |r_x|) \cdot \|z'\| + |r_x| \cdot \|c\| \right).$$

We substitute the simplified bounds from Lemma 4.4:

$$
\begin{aligned}
&\leq \left\| H_\lambda^{-1} \right\| \left( \left( |C|(4 + \|w^*\|\sqrt{d}) + |\beta|(4 + \|w^*\|\sqrt{d}) \right) \sqrt{d} + (4 + \|w^*\|\sqrt{d}) \cdot 4(1 + |\beta|) \right) \\
&= \left\| H_\lambda^{-1} \right\| (4 + \|w^*\|\sqrt{d}) \left( (|C| + |\beta|)\sqrt{d} + 4(1 + |\beta|) \right) \\
&= \left\| H_\lambda^{-1} \right\| (4 + \|w^*\|\sqrt{d}) \left( |C|\sqrt{d} + |\beta|\sqrt{d} + 4 + 4|\beta| \right) \\
&= \left\| H_\lambda^{-1} \right\| (4 + \|w^*\|\sqrt{d}) \left( |C|\sqrt{d} + |\beta|(\sqrt{d} + 4) + 4 \right).
\end{aligned}
$$

This gives the first line of the theorem. For a large feature dimension $d$, we have $|\beta| \approx 1$. The expression's dominant terms are:

$$
\begin{aligned}
&\lesssim \left\| H_\lambda^{-1} \right\| (4 + \|w^*\|\sqrt{d}) \left( |C|\sqrt{d} + 1 \cdot (\sqrt{d} + 4) + 4 \right) \\
&= \left\| H_\lambda^{-1} \right\| (4 + \|w^*\|\sqrt{d}) \left( (|C| + 1)\sqrt{d} + 8 \right) \\
&= \left\| H_\lambda^{-1} \right\| \left( 4(|C| + 1)\sqrt{d} + 32 + (|C| + 1)\|w^*\|d + 8\|w^*\|\sqrt{d} \right).
\end{aligned}
$$

Grouping by powers of $d$, the highest-order terms give the final asymptotic bound:

$$\left\| w_x - w_x' \right\| \lesssim \left\| H_\lambda^{-1} \right\| \left( (|C| + 1)\|w^*\|d + 4(|C| + 1)\sqrt{d} \right).$$

$$\square$$

## B.1 Similarity Measure Evaluation

To validate our choice of Pearson correlation, we empirically evaluated three similarity measures for approximating sequential unlearning effects. Our experimental setup involved training a linear regression model on the Diabetes dataset with L2 regularization ($\lambda = 0.01$) and testing sequential unlearning approximations on 4,950 pairs of randomly subsampled training points.

We compared three measures to calculate the approximation factor $\alpha$:

$$\alpha_{\text{cos}} = \frac{x^\top z'}{\|x\|_2 \|z'\|_2},$$

$$\alpha_{\text{pearson}} = \frac{(x - \bar{x})^\top (z' - \bar{z}')}{\|x - \bar{x}\|_2 \|z' - \bar{z}'\|_2},$$

$$\alpha_{\text{proj}} = \frac{x^\top z'}{\|z'\|_2^2}$$

Each similarity measure captures distinct facets of data relationships. *Cosine similarity* ($\alpha_{\text{cos}}$) measures the angle between two vectors, assessing directional congruence independent of magnitude. The *Pearson correlation coefficient* ($\alpha_{\text{pearson}}$) quantifies the linear correlation between centered variables, making it robust to distributional shifts. Finally, *Projection-based similarity* ($\alpha_{\text{proj}}$) is the unnormalized scalar projection of one vector onto another, making it sensitive to vector magnitudes.

For each pair $(z', x)$, we measured the approximation quality by computing the parameter difference $\|w_x - w_x'\|_2$ between standard approximate unlearning and our similarity-based approach. A similarity measure "wins" if it produces the smallest approximation error among the three methods for that pair.

Table 5: Comparison of similarity measures on the Diabetes dataset across 4,950 unlearning pairs.

| Similarity Measure | Wins | Win Rate (%) |
|---|---|---|
| Cosine Similarity | 1,160 | 23.4 |
| **Pearson Correlation** | **2,139** | **43.2** |
| Projection-based | 1,651 | 33.3 |

Of the 4,950 pairs tested, Pearson correlation achieved the highest win rate at 43.2%, significantly outperforming the other methods, as shown in Table 5. This superior performance validates our choice of Pearson correlation for capturing the linear relationships most relevant to gradient similarity in unlearning contexts. Furthermore, its mean-centering property naturally accounts for the bias terms in linear models, ensuring that the similarity factor $\alpha$ captures the true linear relationship between data points rather than spurious correlations due to offsets.

## C  Safety Analysis

The safety analysis is performed by setting a maximum acceptable approximation error ($\epsilon_{\text{max}}$) for the unlearned parameters ($\|w_x - w_x'\|$) and solving for the minimum required Pearson correlation ($\alpha$).

We use the overall error bound from Theorem 4.1:

$$\|w_x - w_x'\| \ \leq \ \|H_\lambda^{-1}\| \cdot \|\nabla f(x, w^*) - C\nabla f(z', w^*)\| \ \leq \ \epsilon_{\text{max}}$$

$$\|\nabla f(x, w^*) - C\,\nabla f(z', w^*)\| \ \leq \ \frac{\epsilon_{\text{max}}}{\|H_\lambda^{-1}\|}$$

Now define the allowable gradient error, $\epsilon' = \frac{\epsilon_{\text{max}}}{\|H_\lambda^{-1}\|}$

$$\|\nabla f(x, w^*) - C\,\nabla f(z', w^*)\| \ \leq \ \epsilon'$$

Now, we apply the mathematical bound for the gradient error (derived in Lemma 4.2)

$$\|\nabla f(x, w^*) - C\nabla f(z', w^*)\| \ \leq \ (|C| \cdot |r_z| + |\beta| \cdot |r_x|) \cdot \|z'\| \ + \ |r_x| \cdot \|c\|,$$

where $C = \dfrac{\alpha + 1}{1 - s_{z'}^{(\lambda)}}$,    $r_x$ and $r_z$ are the residuals,    and $x = \beta z' + c$ is the linear relationship.

$$(|C| \cdot |r_z| + |\beta| \cdot |r_x|) \cdot \|z'\| + |r_x| \cdot \|c\| \ \leq \ \epsilon'$$

Subtract the non-$C$ error components (residual and offset terms) on both sides:

$$|C| \cdot |r_z| \cdot \|z'\| \ \leq \ \epsilon' - \underbrace{(|\beta| \cdot |r_x| \cdot \|z'\| + |r_x| \cdot \|c\|)}_{\text{Error not controlled by } \alpha}$$

Solve for the maximum allowed magnitude of $C$, denoted $C_{\max}$:

$$|C| \ \leq \ \frac{\epsilon' - (|\beta| \cdot |r_x| \cdot \|z'\| + |r_x| \cdot \|c\|)}{|r_z| \cdot \|z'\|} \ \equiv \ C_{\max}$$

The scaling factor $C$ is defined in the paper as:

$$C = \frac{\alpha + 1}{1 - s_{z'}^{(\lambda)}}.$$

Since we want to find the lowest $\alpha$ that still satisfies the maximum allowed magnitude $C_{\max}$, we substitute $C_{\max}$ back into the definition and solve for minimum Pearson Correlation $\alpha_{\min}$:

$$\frac{\alpha_{\min} + 1}{1 - s_{z'}^{(\lambda)}} = C_{\max},$$

$$\alpha_{\min} = C_{\max} \cdot \left(1 - s_{z'}^{(\lambda)}\right) - 1.$$

Hence, if $\alpha \geq \alpha_{\min}$, the Pearson correlation ($\alpha$) is high enough to ensure that the error introduced by the shortcut ($\|w_x - w_x'\|$) will be less than or equal to the maximum acceptable error ($\epsilon_{\max}$).

