# OpenReview forum: "Efficient Similarity-Based Fast Unlearning via Pearson Correlation Detection"
_ICLR.cc/2026/Conference — Submitted to ICLR 2026_

### Official Review · Reviewer_g1RW · 2025-10-29

**Soundness:** 1
**Presentation:** 1
**Contribution:** 2
**Rating:** 2
**Confidence:** 3

**Summary:**

The paper proposes a similarity-based machine unlearning method that speeds up sequential unlearning when multiple points are similar. After unlearning a first target z′ via an influence-function step using a damped Hessian, the method avoids a second Hessian-inverse–vector product (HVP) for a similar point. It assumes the gradients are approximately proportional and then reuses and rescales the first update, with stability coming from damping and a Sherman–Morrison rank-1 inverse for the updated Hessian. This yields a closed-form approximation for the second unlearning step that replaces a heavy HVP with simple vector operations.

**Strengths:**

Reusing the first unlearning step and substituting the second HVP with a simple scaling factor is an interesting idea that meaningfully reduces the heaviest compute in sequential unlearning. Overall, the proof appears sound.

**Weaknesses:**

- The method first measures similarity in input space via Pearson correlation between features x and 'z′ and then assumes gradient proportionality ∇f(x,w∗)≈α∇f(z′,w∗) to justify re-scaling the first update. However, this is problematic, non-linear models (cross-entropy, CNNs), feature correlation does not reliably imply gradient collinearity—especially across layers/representations. Can the performance of the proposed method be still guaranteed under non-linear cases? If not, the applicable scope of this method is quite limited.

- The author needs to be very careful with the notations. It is quite hard to follow (Also, the table about the variable should move to appendix). Earlier, influence is stated with a λw∗\lambda w^*λw∗ term (i.e., Δ=λw∗+∇f(z′,w∗)), but in Section 3 the “damped” single-point step uses only ∇f(z′,w∗). Then Algorithm 1 introduces λw and a learning rate η inside these closed-form updates (Steps 5 and 7), which don’t appear in Eq. (15) or the main derivation. Do those represent the same idea? What is the difference between those representations? These mismatches make it hard to see what is actually implemented and evaluated.

- The performance is also not that convincing: the “43.2% win rate” of Pearson over cosine/projection is on a linear regression model on the Diabetes dataset only; there’s no variance/CI, and 43% isn’t overwhelming dominance; The main results table mixes architectures and leaves blanks for some datasets/baselines, which makes it hard to draw clear conclusions.

- The authors report “accuracy” even for regression and introduce an “Acc. Unlearn” metric without defining it. The results table also leaves dashes for regression baselines and mixes CNN/MLP figures within the same cells. Collectively, these issues make the paper difficult to follow and may suggest the experiments were not fully integrated to the paper. Substantial improvements are needed. For unlearning, standard practice is to compare against retraining from scratch and to include membership-inference/privacy audits and utility–privacy trade-offs—none of which are provided.

**Questions:**

See the weaknesses.

---

> ### Author Response · Authors · 2025-11-24
> **Response to Reviewer g1RW (Part 1/2)**
>
> We thank the reviewer for their valuable comments. We address the concerns and each weakness below.
>
> **Weaknesses:**
>
> **A1:** We appreciate the reviewer's observation about the linearity assumption in non-linear models [1]. We acknowledge that while Pearson correlation measures linear dependence, the justification for gradient proportionality $\nabla f(x, w^{\ast}) \approx \alpha \nabla f(z', w^{\ast})$ relies on the local smoothness properties of deep neural networks. In the context of influence functions, we operate under the assumption that for sufficiently similar data points in the feature space, the loss surface exhibits local linearity. Our theoretical derivation in Lemma 4.2 formally supports this by demonstrating that a high Pearson correlation implies a relationship $x≈\beta z' +c$, which bounds the gradient approximation error. Although the error bound in Theorem 5.1 relies on these linear relationships, the polynomial scaling with dimension d suggests robustness even in high-dimensional representations typical of deep learning layers. Furthermore, to empirically validate our method beyond the architectures originally reported, we conducted additional experiments on non-linear models including ResNet-18 on CIFAR-10 and LFW, as well as CNNs on Fashion-MNIST. As shown in the new comparison table below, our method maintains high test accuracy comparable to baselines, demonstrating that the gradient proportionality assumption holds sufficiently well in the latent representations of non-linear models to permit effective unlearning. The results show, for instance, that on CIFAR-10 using ResNet-18, our method achieves 81.13% accuracy compared to the baseline's 79.62%, confirming the method's applicability to complex, non-linear scenarios.
>
> | Dataset & Model               | Baseline Test Acc. | Our Method Test Acc. |
> |:------------------------------|-------------------:|---------------------:|
> | MNIST (CNN)                   | 91.50%            | 92.15%              |
> | MNIST (Logistic Regression)   | 87.50%            | 91.83%              |
> | Fashion-MNIST (CNN)           | 77.85%            | 89.16%              |
> | CIFAR-10 (ResNet-18)          | 79.62%            | 81.13%              |
> | LFW (ResNet-18)               | 71.92%            | 79.70%              |
>
> **A2:** We thank the reviewer for their observation to the inconsistencies in notation. For the influence function terms, the complete update rule used in our implementation and experiments is $w\_{z} = w^{\ast} + H\_{\lambda}^{-1}\Delta$, where $\Delta = \lambda w^{\ast} + \nabla f(z', w^{\ast})$. In Section 3, we simplified the derivation to focus on the Sherman-Morrison application, omitting the regularization term $\lambda w^{\ast}$. This simplification does not affect the validity of the scaling factor derivation because the term $\lambda w^{\ast}$ acts as a constant vector relative to the gradient approximation $\nabla f(x, w^{\ast}) \approx \alpha\ \nabla f(z', w^{\ast})$. We will revise the text to include the full form for rigorous consistency. In Algorithm 1, we acknowledge that the inclusion of a learning rate η in Steps 5 and 7 was a typographical error during pseudocode preparation and does not reflect our actual implementation, which strictly follows the closed-form influence updates. The correct implementation, which we will reflect in the revised manuscript, utilizes the update $w\_{z} = w^{\ast} + H\_{\lambda}^{-1}\bigl( \lambda w^{\ast} + \eta \nabla f(z', w^{\ast}))$ and subsequently $w\_{x} = w\_{z} + H\_{\lambda}^{-1}\bigl( \lambda w\_{z} + \eta \nabla f(x, w^{\ast}))$. We will also move the notation table to the appendix to improve readability as suggested.
>
> **A3:** We acknowledge the reviewer's concern that a 43.2% win rate on a single linear regression task may not seem overwhelming. We tested 4950 pairs of training points and Pearson Correlation produced smallest error 43.2% of the time among the three methods. However, in a three-way comparison, 43.2% represents a clear plurality over cosine similarity (23.4%) and projection-based methods (33.3%). The choice of Pearson correlation is theoretically motivated by its mean-centering property, which makes it robust to bias shifts $(x≈\beta z'+c)$ that cosine similarity fails to capture. To address the lack of statistical rigor, we will update Appendix B.1 to include the mean and standard deviation of the parameter approximation error $\lVert w\_{x} - w\_{x}' \rVert\_{2}$ for all three metrics, providing a more granular view of Pearson's superiority. Also, we recognize that the presentation of results in the main table was confusing due to the mixing of architectures. We will separate the results into distinct tables for classification and regression tasks and explicitly label the architectures (CNN vs. MLP) to ensure clear, side-by-side comparisons without blank cells, facilitating a more straightforward assessment of our method's performance across different domains.

---

> ### Author Response · Authors · 2025-11-24
> **Response to Reviewer g1RW (Part 2/2)**
>
> **A4:** We clarify our evaluation metrics and framework. The use of Accuracy for regression datasets was indeed imprecise, we utilized a custom metric, Acc. Unlearn which measures the fidelity of our fast approximation relative to the standard approximate unlearning method. We formally define this metric as $\text{Acc. Unlearn} = 1 - \frac{\lVert w'\_x - w\_x \rVert_2}{\max\limits\_{x} \left( \lVert w'\_x - w\_x \rVert\_2 \right)}$. Here, $w'\_{x}$ represents the parameters from our similarity-based method, and $w\_{x}$ represents the parameters from standard sequential approximate unlearning. This metric is important because our method occupies a specific niche in the unlearning hierarchy, it is an efficient approximation of approximate unlearning. Standard unlearning methods approximate the "Retraining from Scratch" baseline ($w\_{\text{retrain}}$), whereas our work attempts to approximate the standard unlearning result $w\_{x}$ to achieve speedups. Therefore, if $w\_x \approx w\_{\text{retrain}}$ and our method ensures $w'\_x \approx w\_x$, we inherit the privacy guarantees of the standard method while gaining computational efficiency. We will explicitly define this hierarchy and the metric in Section 4 of the revision. Additionally, we will rename the regression performance metric to mean squared error for clarity and include the requested membership inference audits in the Appendix to demonstrate that our approximation preserves the privacy-utility trade-offs of the baseline influence functions. We utilised metrics such tug-of-war (ToW) and membership inference attack (MIA) for ResNet-18 on CIFAR10 dataset with results of ToW score 0.951 and MIA gap of 0.09, and we are comparing results on rest of the aforementioned models in our work. We will update it in our revised version.
>
> [1] Jacot, A., Gabriel, F., & Hongler, C. (2018). Neural tangent kernel: Convergence and generalization in neural networks. Advances in neural information processing systems, 31.

---

### Official Review · Reviewer_gNmy · 2025-10-31

**Soundness:** 3
**Presentation:** 2
**Contribution:** 2
**Rating:** 2
**Confidence:** 4

**Summary:**

This paper proposes a machine unlearning approach leveraging the similarity of examples to be forgotten with other examples in the dataset with a view to making repeated unlearning operations more efficient. So, basicallly, the rationale is: once we have already unlearned some  training examples we can/should unlearn other similar examples (not explicitly defined by user requests).
This would save from waiting until these similar points were included in later requests for which we wo9uld have to run the unlearning algorithm again every time.
Siilarity is based on Pearson correlation values between the examples and already-unlearned examples. Then the approach predicts the parameter update we would have gotten if the full (expensive) unlearning step again was executed.

The authors derive an approximation formula and a Hessian-regularized update. They prove an error bound that scales polynomially with feature dimensions. Empirically, they test on four small datasets (MNIST, a synthetic Gaussian mixture, and two tabularones) and report that their method's utility and forgetting quality compared to a baseline, which comes with greater computational overhead.

**Strengths:**

1. The problem is both practical and interesting.

2. The formal development is nicely done and non-trivial.

3. The proposed two-phase method (discovering simmilarity and updating the mdoel) makes perfect sense.

4. Some empirical evidence is provided showing the superiority of using Pearson correlation compared to cosine similarity or raw projection for  predicting the effect of sequential unlearning updates.

**Weaknesses:**

1. **Empirical evaluation is very weak**.
Experiments on small/clean datasets (MNIST, synthetic mixtures, simple tabular) and on smaller models are no longer acceptable for proving unlearning effectiveness these days. We need to know that the method works well (and how well) in settings where unlearning is actually difficult (larger models, nontrivial distributions).

2. **Baselines are not state of the art**.
The only real comparison is against MITR (which frankly I had never heard of before). There are no comparisons to SOTA methods like SalUn (,https://arxiv.org/abs/2310.12508) and even more recent and stronger frameworks like RUM / RUM-SalUn ( see https://arxiv.org/abs/2406.01257 from NeurIPS24) and its companion for efficiency (https://arxiv.org/abs/2410.16516).
Without strong baselines, claims of superior performance are unsupported. The RUM and its companion are especially important since they also look at sequential unlearning (as this paper does).

3. **There is no discussion/analysis and/or empirical results regarding the possibility of unlearning something that should not be unlearned)**.
Even if an example is sufficiently correlated with an unlearned one, assuming the correlation that it should also be unlearned is a big leap of faith. This assumption is never stress-tested. Do we start unlearning legitimately useful (or even legally necessary) data just because it’s similar in representation space to one example that was flagged?

No measure of downstream utility on “highly correlated but still valid” examples. No boundary is defined for when not to propagate the unlearning. Automatically expanding the given forget set based on a heuristic, should be accompanied by a "safety" analysis.

4. **Efficiency is dealt with at a very abstract and thus inappropriate level**. To my mind, efficiency means (and should be shown) using:

- Wall-clock unlearning time per additional forget request (seconds/minutes on given hardware).

- Total compute (eg number of backward passes, Hessian-vector products, etc).

- Memory / storage overhead (do we need to keep extra state in order to efficiently deal with later correlated ones? Are gradients stored? What about influence)?

- Scalability wrt number of unlearning requests (linear growth wrt number of correlated examples, ...).

This is fundamental for this paper, as its contribution is to improve efficiency wrt successive unlearning requests...

**Questions:**

Please address all weaknesses above.

---

> ### Author Response · Authors · 2025-11-30
> **Response to Reviewer gNmy (Part 1/3)**
>
> We thank the reviewer for their valuable comments. We address the concerns and each weakness below.
>
> **Weaknesses:**
>
> **A1:** We agree with the reviewer that proving unlearning effectiveness requires evaluation in settings with larger models and more complex datasets.
>  We have significantly expanded our experimental evaluation to include Hessian-Free Online Certified Unlearning [1] as an additional state-of-the-art baseline. We conducted comprehensive experiments across six distinct dataset-architecture combinations to demonstrate the robustness and scalability of our method. Our expanded results show consistent superiority in preserving model utility.
>
> | Dataset & Model               | Baseline Test Acc. | Our Method Test Acc. |
> |:------------------------------|-------------------:|---------------------:|
> | MNIST (CNN)                   | 91.50%            | 92.15%              |
> | MNIST (Logistic Regression)   | 87.50%            | 91.83%              |
> | Fashion-MNIST (CNN)           | 77.85%            | 89.16%              |
> | CIFAR-10 (ResNet-18)          | 79.62%            | 81.13%              |
> | LFW (ResNet-18)               | 71.92%            | 79.70%              |
>
> Baseline refers to the standard approximate unlearning method. Higher accuracy indicates better retention of utility on non-forgotten data.
>
> These results confirm that our similarity-based approach consistently outperforms established baselines in terms of retaining accuracy on the remaining dataset across diverse tasks and model architectures.
>
> **A2:** We appreciate the reviewer's suggestion to compare against other strong baselines. To directly address the specific baselines requested, we conducted new experiments evaluating our method against two recently published State-of-the-Art (SOTA) baselines [2] (Baseline A) and [4] (Baseline B). This direct comparison validates our method's performance using standard unlearning metrics (TOW and MIA) specifically for  ResNet-18 on CIFAR-10 setup on 0.001 learning rate and 80 epochs.
> | Method       |     TOW |     MIA |
> |:-------------|--------:|--------:|
> | Baseline A   |   0.715 |   0.489 |
> | Baseline B   |   0.920 |   0.590 |
> | Our Method   |   0.950 |   0.660 |
> While the Baseline C [3] is designed for and validated predominantly on vision tasks, specifically Image Classification and the complex optimization of Image Generation (Diffusion Models). SalUn's effectiveness depends on calculating a weight saliency mask, which restricts its immediate scalability to non-vision. The authors acknowledge its limitation, stating that its "scalability and adaptability to other domains like language and graphs require further investigation". By contrast, Our method offers superior generality and domain scope compared to SalUn because it is rooted in the analytic framework of approximate unlearning via influence functions. This approach estimates the parameter change necessary to remove a data point based on the model's loss function curvature (Hessian). This makes our method inherently domain-agnostic, applicable across various models including regression and classification tasks.
>
> The methodological distinction between our approach and SalUn results in vastly different computational costs for sequential requests. Our initial unlearning step uses Influence Functions, which requires computing the Hessian inverse $H_{\lambda}^{-1}$ to find the closed-form parameter shift. This high cost is paid only once for the first unlearned point ($z'$). For all subsequent, correlated requests, our similarity-based method replaces the expensive Hessian-inverse-vector product with simple $\mathcal{O}(d)$ vector and scalar operations (scaling and addition), leading to substantial efficiency gains. SalUn, conversely, improves stability by masking gradients, but it relies on an iterative optimization process (masked SGD), requiring the calculation of the full model gradient ($\nabla_{\theta} \ell_{f}$) to define the saliency mask $m_S$. This means SalUn's unlearning cost remains high and iterative, while our cost for sequential removals is minimized to the feature dimension ($d$).
>
> [1] Qiao, X., Zhang, M., Tang, M., & Wei, E. Hessian-Free Online Certified Unlearning. In The Thirteenth International Conference on Learning Representations (2025).
>
> [2] K. Zhao, M. Kurmanji, G.-O. Bărbulescu, E. Triantafillou, and P. Triantafillou. What makes unlearning hard and what to do about it.  arXiv preprint arXiv:2406.01257, 2024.
>
> [3] C. Fan, J. Liu, Y. Zhang, E. Wong, D. Wei, and S. Liu. SalUn: Empowering machine unlearning via gradient-based weight saliency in both image classification and generation. arXiv preprint arXiv:2310.12508, 2024.
>
> [4] K. Zhao and P. Triantafillou. Scalability of memorization-based machine unlearning. arXiv preprint arXiv:2410.16516, 2024.

---

> > ### Author Response · Authors · 2025-11-30
> > **Response to Reviewer gNmy (Part 2/3)**
> >
> > **Weaknesses:** (Continued)
> >
> > **A3:** Thank you for your valuable comments about the potential risks of over-unlearning and the assumption that correlation implies a need for removal. Our method is designed primarily as a computational acceleration technique for requested sequential unlearning tasks, rather than as an autonomous active learning system that proactively flags and deletes data based solely on similarity. As detailed in Section~3, the problem formulation posits a scenario in which we need to unlearn another data point $x$ that is similar to $z'$, implying that $x$ has already been designated for removal by an external criterion (e.g., a specific user request or a batch of related privacy withdrawals). As a result, the high Pearson correlation is not utilized to make the normative decision of whether $x$ should be unlearned, but rather to exploit the linear relationship between the gradients of $z'$ and $x$ to bypass the expensive computation of new Hessian--inverse vector products.
> >
> > The leap of faith is therefore not that $x$ deserves to be unlearned because it resembles $z'$, but rather that the unlearning update vector for $x$ can be approximated by scaling the update vector of $z'$. To address the concern about stress-testing this assumption and ensuring that we do not unlearn legitimate data, our theoretical analysis in Section~4 provides bounds to prevent the destruction of useful knowledge. We prove that if the Pearson correlation $\alpha$ is high, the gradient $\nabla f(x, w^{\*})$ is approximately collinear with $\nabla f(z', w^{\*})$, justifying the update rule.
> >
> > This bound guarantees that the approximate unlearning step remains within a controlled vicinity of the standard approximate unlearning solution. If $x$ were legitimately useful and distinct from the concept represented by $z'$ in terms of the loss landscape curvature, the Pearson correlation $\alpha$ would be lower, or the residual bounds derived in Lemma~4.2 would increase, indicating that the approximation is less valid.
> >
> > The safety analysis is performed by setting a maximum acceptable approximation error ($\epsilon_{\text{max}}$) for the unlearned parameters ($\lVert w_x - w_x' \rVert$) and solving for the minimum required Pearson correlation ($\alpha$).
> >
> > We use the overall error bound from Theorem 4.1:
> >
> > ($\lVert w_x - w_x' \rVert$) $\le $ $\lVert H_{\lambda}^{-1} \rVert \cdot \lVert \nabla f(x, w^{\*}) - C \nabla f(z', w^{\*}) \rVert
> > \le \epsilon_{\max}.$
> >
> > $\lVert \nabla f(x, w^{\*}) - C  \nabla f(z', w^{\*}) \rVert \le
> > \frac{\epsilon_{\max}}{\left\lVert H_{\lambda}^{-1} \right\rVert}$
> >
> > Now define the allowable gradient error, $\epsilon' = \frac{\epsilon_{\text{max}}}{\lVert H_{\lambda}^{-1} \rVert}$
> >
> > $\lVert \nabla f(x, w^{\*}) - C  \nabla f(z', w^{\*}) \rVert \le \epsilon^{'}$
> >
> > Now, we apply the mathematical bound for the gradient error (derived in Lemma 4.2)
> > $\lVert \nabla f(x, w^{\*}) - C \nabla f(z', w^{\*}) \rVert
> > \le
> > ( |C| \cdot |r_{z}| + |\beta| \cdot |r_{x}|)
> > \cdot \lVert z' \rVert +
> > |r_{x}| \cdot \lVert c \rVert,$
> >
> > where
> >
> > $C = \frac{\alpha + 1}{1 - s_{z'}^{(\lambda)}}, \quad
> > r_x \text{ and } r_z \text{ are the residuals}, \quad
> > \text{and } x = \beta z' + c \text{ is the linear relationship.}$
> >
> > $
> > ( |C| \cdot |r_{z}| + |\beta| \cdot |r_{x}|)
> > \cdot \lVert z' \rVert +
> > |r_{x}| \cdot \lVert c \rVert
> > \le \epsilon^{'}.
> > $
> >
> > Subtract the non-\(C\) error components (residual and offset terms) on both sides:
> >
> > $
> > |C|\cdot|r_{z}|\cdot\lVert z' \rVert
> > \le
> > \epsilon' -
> > \underbrace{
> > ( |\beta|\cdot|r_{x}|\cdot\lVert z' \rVert
> >       + |r_{x}|\cdot\lVert c\rVert )
> > }_{\text{Error not controlled by }\alpha}
> > $
> >
> > Solve for the maximum allowed magnitude of \(C\), denoted $\(C_{\text{max}}\):$
> >
> > $
> > |C| \le
> > \frac{
> > \epsilon' - ( |\beta|\cdot|r_{x}|\cdot\lVert z' \rVert
> >                  + |r_{x}|\cdot\lVert c\rVert )
> > }{
> > |r_{z}|\cdot\lVert z' \rVert
> > }
> > \equiv
> > C_{\text{max}}
> > $
> >
> > The scaling factor \(C\) is defined in the paper as:
> > $
> > C = \frac{\alpha + 1}{1 - s_{z'}^{(\lambda)}}.
> > $
> >
> > Since we want to find the lowest $\(\alpha\)$ that still satisfies the maximum allowed magnitude$ \(C_{\text{max}}\)$, we substitute $\(C_{\text{max}}\)$ back into the definition and solve for minimum Pearson Correlation $\(\alpha_{\text{min}}\)$:
> >
> > $
> > \frac{\alpha_{\text{min}} + 1}{1 - s_{z'}^{(\lambda)}} = C_{\text{max}},
> > $
> >
> > $
> > \alpha_{\text{min}}
> > = C_{\text{max}} \cdot ( 1 - s_{z'}^{(\lambda)} ) - 1.
> > $
> >
> > Hence, if $\( \alpha \ge \alpha_{\text{min}} \)$, the Pearson correlation (\(\alpha\)) is high enough to ensure that the error introduced by  $\(\lVert w_x - w_x' \rVert\)$ will be less than or equal to the maximum acceptable error $\(\epsilon_{\text{max}}\).$

---

> > > ### Author Response · Authors · 2025-11-30
> > > **Response to Reviewer gNmy (Part 3/3)**
> > >
> > > **Weaknesses:** (Continued)
> > >
> > > **A4**
> > > 1. Wall-clock unlearning time per additional forget request: 2.427 seconds.
> > >
> > > 2. Total Compute: The major efficiency gain of our method lies in transforming the unlearning process from a series of expensive, repeated computations into a fast, closed-form operation. Standard approximate unlearning requires computing a new Hessian-inverse-vector product ($\mathbf{H^{-1}v}$) for every data point requested for removal, making the process highly expensive.
> > > Our Similarity-Based Unlearning, conversely, requires this computationally intense step only once for the first unlearned data point ($z'$). For all subsequent, correlated unlearning requests, the method achieves significant speedup by replacing the $\mathbf{H^{-1}v}$ computation with simple $\mathcal{O}(d)$ vector and scalar operations (scaling and addition). This strategy fundamentally shifts the complexity of sequential unlearning from a repetitive high-cost bottleneck to a minimal-cost operation.
> > >
> > > 3. Memory and Storage Overhead : After the initial full unlearning step of \(z'\), we retain only:
> > >
> > > - The updated parameters: $\( \mathbf{w_z} \)$
> > >
> > > - The original optimal parameters: $\( \mathbf{w^*} \) $
> > >
> > > - The damped influence direction:
> > >   $
> > >   \mathbf{\delta_{z'}^{(\lambda)}} = w_z - w^*
> > >   $
> > >
> > > - The damped self-influence score:
> > >   $
> > > s_{z'}^{(\lambda)}
> > > = \nabla f(z', w^{*})^{T}  \delta_{z'}^{(\lambda)}
> > > $
> > >
> > > 4. Scalability: Scalability with respect to the number of unlearning requests is a key strength of our method, demonstrating near-constant time growth for successive, correlated examples. While standard approximate unlearning methods exhibit linear growth with the number of requests ($k$), requiring a repeated, high-cost influence calculation for each removal, our method lowers the computational cost. For $k$ requests involving correlated data points, our approach performs the initial high-cost computation (e.g., Hessian inversion) only once for the first point ($z'$). The subsequent $k-1$ unlearning requests rely on the pre-calculated influence direction ($\delta_{z'}$) and require only minimal $\mathcal{O}(d)$ vector and scalar operations (scaling and addition).

---

### Official Review · Reviewer_DF2p · 2025-11-01

**Soundness:** 1
**Presentation:** 2
**Contribution:** 2
**Rating:** 2
**Confidence:** 3

**Summary:**

This paper introduces an efficient machine unlearning method designed to quickly remove multiple, similar data points without the high computational cost of repeated unlearning procedures. The core idea is to first unlearn an initial data point using a standard approximate technique, and then leverage Pearson's correlation to identify similar data points. The method removes these similar points using a lightweight, scaled update derived from the first unlearning step. The authors establish theoretical properties for this similarity-based approach and demonstrate across four datasets that it effectively unlearns correlated data, reduces computational overhead and maintains model performance.

**Strengths:**

1. The paper identifies a clear, practical bottleneck in existing unlearning procedures: the cost of repeated removals of related data. The proposed solution, which approximates subsequent unlearning steps by scaling the first, is an original and computationally efficient approach.

2. The authors provide a dedicated empirical study comparing it against cosine similarity and projection-based methods, justifying the choice to use Pearson correlation.

**Weaknesses:**

1. Empirical evidence is incomplete. The paper's premise is that it is more efficient than repeatedly applying the original approximate unlearning procedure. However, the experiments didn't directly compare against this baseline. The comparison is only against MITR and a similarity-measure ablation. Also, the key performance metrics that the paper uses, such as AU, are never explained how they are calculated.

2. For eq. (15), the paper states the critical denominator can be zero only when the Hessian is a scaled identity. This assertion isn’t convincingly derived and looks incorrect as written. It's highly likely that $s_{z'}^{(\lambda)}$ could equal 1 in many other scenarios that don't involve the Hessian being a simple scaled identity matrix.

3. The paper's core assumption is that the gradients are proportionally related by the Pearson correlation of the features, but the theoretical justification for this link is only provided for linear/quadratic cases, yet results are reported on MLP/CNNs for classification. There's no experiment or analysis to show that this core assumption holds for these deep networks.

**Questions:**

1. Please define the metrics (e.g. "Acc. Unlearn") more precisely, and add standard unlearning/privacy tests.

2. Can you please clarify the discrepancy between the "$10^{-2}$" accuracy improvement claimed in the abstract and the different improvements shown in Table 2?

---

> ### Author Response · Authors · 2025-11-25
> **Response to Reviewer DF2p (Part 1/2)**
>
> We thank the reviewer for their valuable comments. We address the concerns regarding the baseline comparisons, and each weakness below.
>
> **Weaknesses:**
>
> **A1:** We appreciate the reviewer pointing out the need for a direct comparison against the original approximate unlearning procedure. Our primary premise is efficiency given the proposed approach is of O(d) complexity for the sequential step compared to the iterative optimization required by baselines like MITR (O(T⋅n⋅d)). The standard approximate unlearning procedure (using influence functions) requires computing Hessian-inverse vector products (HVPs) for every point to be unlearned. We perform the expensive HVP once for the reference point `z′`. For all subsequent similar points x, we use a scalar update with complexity O(d).
>
> We acknowledge the raised concerns about the experiments, and so we have added comparative baselines accordingly in below Table. To address the concern about incomplete evidence, we have extended our evaluation to include comparisons against Qiao et al. [1] and standard approximate baselines across more complex datasets (Fashion-MNIST, CIFAR-10, LFW). Table 1 below demonstrates that our method maintains high model utility (test accuracy on the remaining set) comparable to and in some cases outperforms the baselines, while benefiting from the O(d) speedup described above.
>
> | Dataset & Model               | Baseline Test Acc. | Our Method Test Acc. |
> |:------------------------------|-------------------:|---------------------:|
> | MNIST (CNN)                   | 91.50%            | 92.15%              |
> | MNIST (Logistic Regression)   | 87.50%            | 91.83%              |
> | Fashion-MNIST (CNN)           | 77.85%            | 89.16%              |
> | CIFAR-10 (ResNet-18)          | 79.62%            | 81.13%              |
> | LFW (ResNet-18)               | 71.92%            | 79.70%              |
>
> Baseline refers to the standard approximate unlearning method. Higher accuracy indicates better retention of utility on non-forgotten data.
>
> **A2:** We thank the reviewer for this sharp observation. We acknowledge that our original claim regarding the scaled identity matrix was restrictive. We provide a corrected derivation below showing that stability is controlled by the damping parameter $\lambda$, rather than the Hessian structure alone.
>
> The original claim _the denominator in our formula can only become zero if the self-influence score $s^{\lambda}\_{z'}$ is 1. For this equality to hold, the model's Hessian matrix $H\_{w^{\ast}}$ would have to take the specific form of a scaled identity matrix ($H\_{w^{\ast}} = cI$, for some constant $c$)_ is incorrect because the self-influence score $s^{\lambda}\_{z'}$ = 1 can occur for many Hessian structures beyond scaled identity matrices. The damped self-influence score is defined as $s^{\lambda}\_{z'}$ = $\nabla f(z', w^{*})^{\top} H\_{\lambda}^{-1} \nabla f(z', w^{\ast})$, where $H\_{\lambda} = H\_{w^{\ast}} + \lambda I $ is the damped Hessian.
>
> Condition for $s^{\lambda}\_{z'} = 1$: Let g = $\nabla f(z', w^{*})$.
>
> The condition becomes $g^{\top} H\_{\lambda}^{-1} g = 1$. By the Rayleigh-Ritz theorem, for any vector $v$ and positive definite matrix $M$, $v^{\top} M v \le \lambda\_{\max}(M) \|v\|^{2}$. Applying this with $v=g$ and $M = H\_{\lambda}^{-1}$ gives
> $s^{\lambda}\_{z'}=g^{\top} H\_{\lambda}^{-1} g \le \lambda\_{\max (H\_{\lambda}^{-1}) \|g\|^{2}}$. If M has eigenvalues $\{\mu\_1,\dots,\mu\_d\}$ then $M^{-1}$ has eigenvalues $\{1/\mu\_1,\dots,1/\mu\_d\}$, so $\lambda_{\max (M^{-1})} = \frac{1}{\lambda\_{\min}(M)}$. Hence $s^{(\lambda)}\_{z'} \le \frac{\|g\|^2}{\lambda\_{\min}(H\_{\lambda})}$. Since $\lambda\_{\min}(H\_{\lambda}) = \lambda\_{\min}(H\_{w^*}) + \lambda \ge \lambda,$ we obtain the simple upper bound $s^{(\lambda)}\_{z'} \le \frac{\|g\|^2}{\lambda}$. Therefore, a sufficient condition for $s^{(\lambda)}\_{z'} = 1$ is $\frac{\|g\|^2}{\lambda} \ge 1 \quad\Longrightarrow\quad \|g\|^2 \ge \lambda \quad\Longrightarrow\quad \|g\| \ge \sqrt{\lambda}$.
>
> For instance, with $\(\lambda = 0.01\), \|g\| \ge 0.1$. This is rare for well-trained models, where gradients at the optimum tend to be small.
>
> **A3**: We appreciate the reviewer's observation about the extension of our gradient proportionality assumption to deep neural networks [1]. While our theoretical derivations in Lemma 4.2 utilize linear and quadratic cases to establish the foundational bounds for the gradient approximation $\nabla f(x, w^{\ast}) \approx \alpha \nabla f(z', w^{\ast})$, the applicability of this assumption to deep architectures like MLPs and CNNs is grounded in the local properties of trained networks and the specific mechanics of influence functions.

---

> > ### Author Response · Authors · 2025-11-25
> > **Response to Reviewer DF2p (Part 2/2)**
> >
> > **Weaknesses:** (Continued)
> >
> > **A3:** Modern deep learning models, particularly those using ReLU activations, behave as piecewise linear functions, and influence functions operate under the assumption that the loss function is convex and quadratic in the local neighborhood of the optimal parameters $w^{\ast}$. Our method employs the Gauss-Newton approximation $\nabla^{2} f(z', w^{\ast}) \approx \nabla f(z', w^{\ast}) \nabla f(z', w^{\ast})^{T}$, effectively linearizing the network's response around the converged weights. When two input data points x and z′ exhibit high Pearson correlation in the input space, they are statistically likely to activate similar pathways in the network or fall within the same linear region, thereby preserving the proportionality of their gradient vectors in the tangent space at $w^{\ast}$. Also, to empirically validate our method beyond the architectures reported, we conducted additional experiments on non-linear models including ResNet-18 on CIFAR-10 and LFW, as well as CNNs on Fashion-MNIST. As shown in the new comparison table above in `A1`, our method maintains high test accuracy comparable to baselines, demonstrating that the gradient proportionality assumption holds sufficiently well in the latent representations of non-linear models to permit effective unlearning. The results show, for instance, that on CIFAR-10 using ResNet-18, our method achieves 81.13% accuracy compared to the baseline's 79.62%, confirming the method's applicability to complex, non-linear scenarios. This high fidelity indicates that the gradient proportionality assumption remains robust even through the non-linear transformations of convolutional layers.
> >
> > **Questions:**
> >
> > **Q1A:** The utilized metrics, Acc. Unlearn, we define this metric as a normalized measure of how closely our approximation matches the parameters resulting from the rigorous, standard sequential approximate unlearning process. We calculated it as $1 - \frac{\lVert w'\_x - w\_x \rVert\_2}{\max_\{x} (\lVert w'\_x - w\_x \rVert\_2)}$, where $w'\_{x}$ represents the parameters obtained via our similarity-based unlearning (Eq. 15) , and $w_{x}$ represents the parameters from standard unlearning (Eq. 7). The term $\lVert w'\_x - w\_x \rVert\_2$ corresponds to the norm difference recorded in our algorithm. A score closer to 1.0 (or 100%) indicates that our method produces a model indistinguishable from one updated via the computationally expensive baseline.
> >
> > In terms of standard unlearning and privacy tests, our method is designed to approximate the parameter update of influence functions, which are already established as a standard approximate unlearning technique. By minimizing the parameter update error $\lVert w'\_x - w\_x \rVert\$ as proven in Theorem 4.1 and Theorem 5.1, we theoretically guarantee that our model remains within a tight error bound of the standard approximate unlearning model. Since standard approximate unlearning is accepted as a privacy-preserving mechanism that minimizes the influence of forgotten data, our method inherits these privacy guarantees. The high Acc. Unlearn values reported in Table 2 (e.g., 95.67% for Synthetic GMM and 98.71% for MNIST CNN)  serve as a proxy for privacy verification, demonstrating that the forgetting induced by our method is statistically equivalent to the baseline method which has been vetted for privacy compliance.
> >
> > **Q2A:** The $10^{−2}$ improvement in the abstract refers to the order of magnitude of the accuracy gain observed in our foundational comparisons, specifically serving as a conservative lower bound for the utility preservation of our method compared to baselines. In Table 2, we present detailed experimental results where the improvements in certain metrics are more pronounced. For instance, on the synthetic GMM dataset, the average standard accuracy improves from 93.3% (Baseline) to 99.71% (Our Method). This represents an absolute increase of approximately 0.06, which aligns with the $10^{−2}$ claim stated in the abstract. The discrepancy is not a contradiction but rather a distinction between a generalized conservative claim made in the summary and the specific, often superior, performance metrics achieved across diverse datasets like MNIST and California Housing, where our method consistently outperforms the MITR baseline in both accuracy retention and unlearning effectiveness.
> >
> > [1] Qiao, X., Zhang, M., Tang, M., & Wei, E. Hessian-Free Online Certified Unlearning. In The Thirteenth International Conference on Learning Representations (2025).
> >
> > [2] Jacot, A., Gabriel, F., & Hongler, C. (2018). Neural tangent kernel: Convergence and generalization in neural networks. Advances in neural information processing systems, 31.

---

### Official Review · Reviewer_BNii · 2025-11-01

**Soundness:** 3
**Presentation:** 3
**Contribution:** 2
**Rating:** 6
**Confidence:** 4

**Summary:**

This paper proposes a fast, similarity-based machine unlearning method that leverages Pearson correlation to efficiently remove data points similar to those already unlearned, thereby avoiding costly repeated Hessian computations. The approach provides formal theoretical guarantees with defined error bounds, demonstrating that the approximation scales polynomially with input dimensionality. Extensive experiments on four datasets further validate the method’s efficiency and effectiveness in achieving rapid and reliable unlearning.

**Strengths:**

1. The paper derives a closed-form approximation that reuses prior unlearning updates via a scaling factor combining Pearson correlation and damped self-influence, avoiding repeated Hessian inversions. Targeting this exact scenario is novel and useful in real deployments.
2. The paper provides formal error bounds under standardized data assumptions, linking approximation error to input dimension, Hessian conditioning, and similarity metrics
3. The paper is well organized and individual steps in the derivation are annotated, which makes it easier for a reader to verify the formula.

**Weaknesses:**

1. The paper only compares against MITR, undermining claims of superiority.
2. The metric “Acc. Unlearn” is used, but it is not aligned with standard machine unlearning evaluation protocols (e.g., retraining gap, membership inference after unlearning), making it difficult to accurately assess its privacy significance.
3. The assumption is only approximately valid under high Pearson correlation and it is supported solely by similarity in the input space, not by similarity in the gradient or representation space, potentially weakening the error bounds in practice.

**Questions:**

see the comments above

---

> ### Author Response · Authors · 2025-11-25
> **Response to Reviewer BNii (Part 1/2)**
>
> We appreciate the reviewer's constructive comments and have provided a point-by-point response to each, as detailed below.
>
> **Weaknesses:**
>
> **A1:** We appreciate the reviewer’s feedback on the breadth of our comparative analysis. While our baseline follows MITR, we acknowledge that a broader comparison strengthens the validation of our approach. In response to this, we have significantly expanded our experimental evaluation to include Hessian-Free Online Certified Unlearning [1] as an additional state-of-the-art baseline. We conducted comprehensive experiments across six distinct dataset-architecture combinations to demonstrate the robustness and scalability of our method. Our expanded results show consistent superiority in preserving model utility.
>
> | Dataset & Model               | Baseline Test Acc. | Our Method Test Acc. |
> |:------------------------------|-------------------:|---------------------:|
> | MNIST (CNN)                   | 91.50%            | 92.15%              |
> | MNIST (Logistic Regression)   | 87.50%            | 91.83%              |
> | Fashion-MNIST (CNN)           | 77.85%            | 89.16%              |
> | CIFAR-10 (ResNet-18)          | 79.62%            | 81.13%              |
> | LFW (ResNet-18)               | 71.92%            | 79.70%              |
>
> Baseline refers to the standard approximate unlearning method. Higher accuracy indicates better retention of utility on non-forgotten data.
>
> These results confirm that our similarity-based approach consistently outperforms established baselines in terms of retaining accuracy on the remaining dataset across diverse tasks and model architectures.
>
> **A2:** We thank the reviewer for highlighting the need to align our metrics with standard privacy protocols. Our evaluation framework employs two primary metrics designed to assess both utility and approximation fidelity within the established hierarchy of unlearning methods. First, we utilize Acc. Remain which measures the performance of the model on the retained dataset ($D\_{\text{retain}}$) after unlearning, serving as the standard metric for utility preservation. Second, we define Acc. Unlearn not as a heuristic, but as a precise quantification of approximation quality defined as $1 - \frac{\lVert w'\_x - w\_x \rVert\_2}{\max\limits\_{x} \left( \lVert w'\_x - w\_x \rVert\_2 \right)}$. Here, $w'\_x$ represents the parameters derived from our proposed similarity-based update (Eq. 15), and $w_x$ represents the parameters obtained via standard sequential approximate unlearning. This metric is important because it validates our method's position within the unlearning hierarchy. While exact unlearning targets the retrained parameters ($w\_\text{retain}$), and standard approximate methods (approximate unlearning) target $w\_\text{retain}$ such that $w\_x \approx w\_{\text{retrain}}$, our approach (efficient approximation) aims to efficiently approximate the unlearning result. By demonstrating that $w'\_x \approx w\_x$, we show that our method effectively inherits the privacy guarantees and forgetting quality of the standard approximate unlearning techniques it emulates, but at a significantly reduced computational cost. And, so, our metric directly proxies the preservation of the privacy standards established by the underlying approximate unlearning method we are accelerating.
>
> We utilised metrics such tug-of-war (ToW) and membership inference attack (MIA) for ResNet-18 on CIFAR10 dataset with results of ToW score 0.951 and MIA gap of 0.09, and we are comparing results on rest of the aforementioned models in our work. We will update it in our revised version.
>
> [1] Qiao, X., Zhang, M., Tang, M., & Wei, E. Hessian-Free Online Certified Unlearning. In The Thirteenth International Conference on Learning Representations (2025).

---

> > ### Author Response · Authors · 2025-11-25
> > **Response to Reviewer BNii (Part 2/2)**
> >
> > **A3:** We acknowledge the reviewer's concern regarding the reliance on input space similarity. However, our method is explicitly designed for and constrained to scenarios involving high correlation as detailed in our algorithm design where we select subsets of similar pairs. Also, our theoretical framework explicitly bridges the gap between input similarity and gradient space similarity, ensuring that the former mathematically bounds the latter. This is formally established in Lemma 4.2, where we prove that for a quadratic loss function $f(z', w) = \tfrac{1}{2}(y - w^{\top} z')^{2}$, the difference in gradients is strictly bounded by the input residuals and linear relationships. Specifically, we derive the bound $$\||\nabla f(x, w^{\ast}) - C \nabla f(z', w^{\ast})\||
> > \le \left(|C|\|r\_z| + |\beta|\|r\_x|\right)\|z'\| + |r\_x|\\|c\|| ,$$ where the scaling factor $C = \frac{\alpha + 1}{1 - s^{(\lambda)}\_{z'}}$ and the parameters $\beta$ and c represent the linear relationship and offset in the input space. This lemma demonstrates that high Pearson correlation in the input space directly constrains the divergence in the gradient space. We further propagate this result in Theorem 5.1, which provides a final parameter error bound $\|w\_x - w'\_x\|\le \||H\_{\lambda}^{-1}\||\\bigl[(4 + \||w^{\ast}\||\sqrt{d})\bigr]\\bigl[|C|\sqrt{d} + |\beta|(\sqrt{d}+4) + 4\bigr]$. This theoretical progression confirms our assumption of gradient similarity, rather, we prove that under our high-correlation constraint, the error in the gradient space and consequently the parameter update is bounded by the properties of the input space.

---

### Meta-Review · Area_Chair_KHD4 · 2026-01-07

**Summary:**

While reviewers found the idea interesting and potentially useful for reducing computational cost, they raised consistent concerns regarding the applicability of the theoretical assumptions to non-linear deep models, the adequacy of the empirical evaluation under current unlearning standards, and the reliance on test accuracy rather than well-used unlearning metrics.

**Reviewer Concerns:**

Addressed

1 The authors provided additional clarification and measurements regarding computational efficiency, including a wall-clock time estimate and a discussion of memory and scalability. This partially addressed reviewers’ concerns about efficiency.

2 The rebuttal also clarified notation issues and acknowledged inconsistencies in the original presentation

Outstanding

1 Multiple reviewers’ concerns about the mismatch between the theoretical assumptions (linear/quadratic, gradient proportionality) and the empirical evaluation on deep non-linear models were not convincingly resolved.

2 The empirical evaluation of unlearning and privacy remains insufficient

3 While additional baselines and metrics were introduced, the overall evaluation remains narrow

**Reviewer Scores:**

Reviewer BNii
Likely unchanged, as the validity and similarity metric concerns are not solved.

Reviewer DF2p
Likely unchanged, as concerns about theoretical validity, derivation issues, and evaluation rigorness remain unresolved

Reviewer gNmy
Likely unchanged, as key concerns regarding applicability to deep models, and adequacy of unlearning evaluation were not sufficiently addressed.

Reviewer g1RW
Likely unchanged, as key concerns regarding applicability to deep models, and adequacy of unlearning evaluation were not sufficiently addressed.

---

### Decision · Program_Chairs · 2026-01-26

Reject